# THE POTENTIAL OF SECOND-ORDER OPTIMIZATION FOR LLMS: A STUDY WITH FULL GAUSS-NEWTON

**Natalie Abreu**
Kempner Institute, Harvard University
natalieabreu@g.harvard.edu

**Nikhil Vyas**[*]
Department of Computer Science, Harvard University
vyasnikhil96@gmail.com

**Sham Kakade**
Kempner Institute, Harvard University
sham@seas.harvard.edu

**Depen Morwani**
Kempner Institute, Harvard University
dmorwani@g.harvard.edu

## ABSTRACT

Recent efforts to accelerate LLM pretraining have focused on computationally-efficient approximations that exploit second-order structure. This raises a key question for large-scale training: how much performance is forfeited by these approximations? To probe this question, we establish a practical upper bound on iteration complexity by applying full Gauss-Newton (GN) preconditioning to transformer models of up to 150M parameters. Our experiments show that full GN updates yield substantial gains over existing optimizers, achieving a 5.4x reduction in training iterations compared to strong baselines like SOAP and Muon. Furthermore, we find that a precise layerwise GN preconditioner, which ignores cross-layer information, nearly matches the performance of the full GN method. Collectively, our results suggest: (1) the GN approximation is highly effective for preconditioning, implying higher-order loss terms may not be critical for convergence speed; (2) the layerwise Hessian structure contains sufficient information to achieve most of these potential gains; and (3) a significant performance gap exists between current approximate methods and an idealized layerwise oracle.
Code is available at:
https://github.com/natalieabreu/full-gauss-newton.

## 1 INTRODUCTION

With rising compute requirements for training large language models (LLMs), improving optimization methods has become a central strategy for improving training efficiency. Better optimizers can directly reduce the *serial runtime* to train an LLM, which is crucial for large-scale models that train from days to months. Optimization for LLMs has traditionally leveraged first-order methods such as SGD and Adam (Kingma & Ba, 2017). However, recent research in optimization has started exploring the use of second-order optimizers for large-scale models, motivated by the faster convergence rates known from theory (Nesterov, 2018) and potential to scale to larger batch sizes (Zhang et al., 2019) – two ways of reducing serial runtime.

Some recent popular second-order methods include Shampoo (Gupta et al., 2018), SOAP (Vyas et al., 2025) and Muon (Jordan et al., 2024b). Shampoo won the recent optimization algorithms benchmark called AlgoPerf (Kasimbeg et al., 2025), outperforming Adam by a margin of 28%. SOAP, a recent generalization of the Shampoo algorithm, has shown impressive performance on language modeling benchmarks, and has been used for training physics-informed neural networks (PINNs) (Wang et al., 2025). Muon has been extensively optimized on the nanoGPT benchmark (Jordan et al., 2024a), and was also recently scaled up to 16B LLMs, showing 50% improvements over AdamW (Liu et al., 2025).

However, these methods do not use complete second-order information, instead focusing on memory- and computationally-efficient approximations of the Hessian. Indeed, precisely storing or

---

[*]Currently at OpenAI

computing the Hessian required for second-order methods such as Newton's method is prohibitively expensive for modern LLMs that have billions of parameters. To remain practical, these methods leverage computationally-efficient estimators for the *layerwise* Hessian of neural networks.

The success of current methods motivates a better understanding of the fundamental potential of second-order optimizers. Our work is driven by the following question:

> *What are the fundamental performance limits of second-order optimization for LLMs, and what structural properties of the Hessian are essential for achieving them?*

To answer this, we first establish the performance limits of an idealized second-order method, full Gauss-Newton (GN), and measure its performance in terms of iteration complexity (the number of steps to reach a target loss). This serves as a practical lower bound for any second-order approach. We also analyze how this idealized method affects the critical batch size (McCandlish et al., 2018; Shallue et al., 2019; Jain et al., 2018b), a key measure of data parallelism efficiency.

To isolate the essential structural properties of the Hessian, we then compare the full GN method to two variants:

1. A prox-linear version of Gauss-Newton (GN-prox-linear) (Burke, 1985; Drusvyatskiy, 2017), which utilizes the higher order information of the loss function itself (see Algorithms 1 and 2 for comparison).

2. A purely layerwise version, which ignores all cross-layer curvature information.

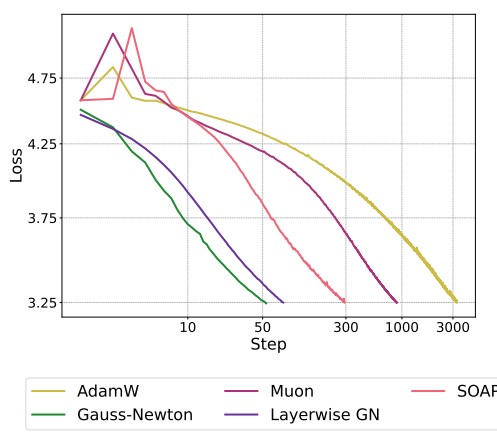

Figure 1: Training step versus validation loss until loss 3.25 when each method is beyond its critical batch size. Gauss-Newton and Layerwise Gauss-Newton reach the target loss in 54 and 78 steps respectively, compared to 292 steps for SOAP.

Our main findings are three-fold. First, both idealized second-order methods provide a substantial improvement over existing optimizers at large batch sizes, with the full Gauss-Newton method achieving a 5.4x reduction in iteration complexity over the SOAP optimizer. Second, the Gauss-Newton method significantly extends the critical batch size beyond that of prior methods, displaying near-optimal scaling. Finally, we find that the layerwise approach, despite its structural limitations, still substantially outperforms both SOAP and Adam. This suggests that even layer-wise curvature information alone is sufficient to achieve major gains in compute efficiency.

We stress that this work is an empirical study aimed at understanding performance limits with better higher order information, not at directly designing computationally cheaper optimizers themselves. Our implementation avoids materializing the Hessian by using Jacobian-vector products, though still has substantial computational overhead. We believe this work is best viewed as a tool for analysis that provides a target for more practical second-order methods to aspire to, and we provide further discussion on this point.

**Paper organization** In Section 2, we cover related work and in Section 3 we provide background on existing optimization methods. In Section 4, we introduce the setting for full second-order optimization and the Gauss-Newton matrix. In Section 5 we provide the setup for our main experiments and in Section 6.1 we discuss our results on iteration complexity and critical batch size of the full second-order method. In Section 6.2 we compare the Gauss-Newton method to the GN-prox-linear method and in Section 6.3 we compare to a layerwise variation. Finally, in Section 7 we discuss the implications as well as limitations of our work.

---

**Algorithm 1:** Gauss-Newton method

---

**Input:** $\theta_0$, training set $\mathcal{T}$, training iterations $T$, batch size $b$
**for** $t = 0, 1, \ldots, T-1$ **do**

> **Linearize model:**
> $f_{\theta_t}^{(1)}(\theta, x) := f(\theta_t, x) + \nabla f(\theta_t, x)^\top (\theta - \theta_t)$
> **Convexify loss:**
> With $B \sim \mathcal{T}$, $\widetilde{\mathcal{L}}_{\theta_t}(\theta) := \frac{1}{b} \sum_{(x,y) \in B} \ell(f_{\theta_t}^{(1)}(\theta, x), y)$
> **Gauss-Newton loss:**
> $\widetilde{\mathcal{L}}_{\theta_t}^{(2)}(\theta) := \widetilde{\mathcal{L}}_{\theta_t}(\theta_t) + \nabla \widetilde{\mathcal{L}}_{\theta_t}(\theta_t)(\theta - \theta_t) + \frac{1}{2}(\theta - \theta_t)^\top \nabla^2 \widetilde{\mathcal{L}}_{\theta_t}(\theta_t)(\theta - \theta_t)$
> **Set $\widehat{\theta}$ to be an approximate solution to the *least squares* problem:**
> $\min_\theta \widetilde{\mathcal{L}}_{\theta_t}^{(2)}(\theta)$
> **Line search:** Set $\theta_{t+1} \leftarrow \theta_t + \alpha^\star(\widehat{\theta} - \theta_t)$, where:
> $\alpha^* \leftarrow \arg\min_\alpha \mathcal{L}(\theta_t + \alpha(\widehat{\theta} - \theta_t))$

---

**Algorithm 2:** GN-prox-linear

---

**Input:** $\theta_0$, training set $\mathcal{T}$, training iterations $T$, batch size $b$
**for** $t = 0, 1, \ldots, T-1$ **do**

> **Linearize model:**
> $f_{\theta_t}^{(1)}(\theta, x) := f(\theta_t, x) + \nabla f(\theta_t, x)^\top (\theta - \theta_t)$
> **Convexify loss:**
> With $B \sim \mathcal{T}$, $\widetilde{\mathcal{L}}_{\theta_t}(\theta) := \frac{1}{b} \sum_{(x,y) \in B} \ell(f_{\theta_t}^{(1)}(\theta, x), y)$
> **Set $\theta_{t+1}$ to be an approximate solution to the *convex* problem:**
> $\min_\theta \widetilde{\mathcal{L}}_{\theta_t}(\theta)$

---

## 2 RELATED WORK

We mention a few of the most related works here and provide additional related work in Appendix C; work on specific optimizers for LLMs is discussed in Section 3. Most related to our work is Hessian-free optimization, which avoids explicit Hessian formation by leveraging Hessian-vector products (Martens, 2010). This approach serves as an alternative to layerwise approximation methods of the Hessian as discussed in Section 3. Specifically, prior work on Hessian-free optimizers use the conjugate gradient (CG) to solve an incomplete (unconverged) optimization of the Newton step rather than storing an approximation to the Hessian. This is introduced by Martens (2010) on classification and auto-encoder tasks, and extended to additional settings such as recurrent neural networks by Martens & Sutskever (2011a) and Cho et al. (2015). Garcia et al. (2023) amortizes the CG steps in Hessian-free optimization for deep linear and auto-encoder models. In contrast, our work focuses on the setting of LLMs, and we leverage optimizers that are specifically designed for LLMs (e.g. Adam and Muon) rather than CG to apply the Gauss-Newton step.

## 3 BACKGROUND ON EXISTING OPTIMIZERS

We will denote the weight matrix of a model layer at timestep $t$ by $W_t \in \mathbb{R}^{m \times n}$ and the corresponding gradient by $G_t$. We use $\eta$ to denote the learning rate.

The most widely used optimizer for LLMs is Adam (Kingma & Ba, 2017). Adam maintains matrices for the first and second moment of the gradient $G_t$, denoted $M_t$ and $V_t$ respectively. Adam performs the element-wise update

$$W_{t+1} := W_t - \eta \frac{M_t}{\sqrt{V_t}}$$

AdaGrad (Duchi et al., 2011) maintains an accumulator over the vectorized gradient $g_t = \text{vec}(G_t) \in \mathbb{R}^{mn}$. The preconditioner $H_t$ and vectorized weights $w_t$ at timestep $t$ are updated as

$$H_t := H_{t-1} + g_t g_t^\top; \quad w_t := w_{t-1} - \eta H_t^{-1/2} g_t$$

Shampoo (Gupta et al., 2018) was originally motivated by AdaGrad, but can be viewed as an approximation of the Gauss-Newton component of the Hessian (Anil et al., 2021; Osawa et al., 2023; Morwani et al., 2024). These methods leverage computationally efficient approximations of the layerwise Hessian to precondition the gradient update. Shampoo maintains a separate preconditioner for each dimension of the weight matrix: For weight matrix $W \in \mathbb{R}^{m \times n}$, Shampoo maintains a left matrix $L_t \in \mathbb{R}^{m \times m}$ and a right matrix $R_t \in \mathbb{R}^{n \times n}$. The update rule is as follows:

$$L_t := L_{t-1} + G_t G_t^\top; \quad R_t := R_{t-1} + G_t^\top G_t; \quad W_t := W_{t-1} - \eta\, L_t^{-1/4} G_t R_t^{-1/4}$$

SOAP (Vyas et al., 2025) is a recent variant of Shampoo that runs AdamW (Loshchilov & Hutter, 2019) in the eigenbasis provided by Shampoo.

Muon (Jordan et al., 2024b) tracks the first moment of the gradient, denoted $M_t$, and performs an orthonormal update:

$$O_t := NS(M_t) \quad W_{t+1} := W_t - \eta O_t$$

where $NS$ denotes a Newton-Schulz orthogonalization procedure. (See Jordan et al. (2024b) for further description of the Newton-Schulz method). Muon without momentum can be seen as a version of Shampoo without preconditioner accumulation (Bernstein & Newhouse, 2024).

These second-order methods[1] have been shown to scale effectively to larger batch sizes (Zhang et al., 2019; Vyas et al., 2025). However, these are restricted by the need for computationally efficient per-layer preconditioners given the high computational and memory requirements for computing the full Gauss-Newton matrix.

## 4    FULL SECOND-ORDER OPTIMIZATION

### 4.1    NOTATION

Let $f(\theta, x)$ denote the model with parameters $\theta$ and input $x$. Let $\mathcal{L}(f(\theta, x), y)$ denote the (convex) loss function which takes the model output and the true labels $y$. We will use either $\nabla_\theta \mathcal{L}$ or $g$ or simply $\nabla \mathcal{L}$ to denote the gradient with respect to $\theta$, $\nabla_f$ to denote the derivative of $\mathcal{L}$ with respect to $f$ and $H := \nabla_\theta^2 \mathcal{L}$ to denote the Hessian. We will use $f_{\theta_t}^{(1)}(\theta, x)$ and $\mathcal{L}_{\theta_t}^{(2)}(\theta) := \mathcal{L}_{\theta_t}^{(2)}(f(\theta, x), y)$ to denote the first-order Taylor expansion of $f$ around $\theta_t$ and the second-order Taylor expansion of $\mathcal{L}$ around $\theta_t$ respectively. Similarly, we will also use $\mathcal{L}(\theta) := \mathcal{L}(f(\theta, x), y)$ when $f$, $x$, and $y$ are clear from context. For simplicity, we will assume that we are working with cross-entropy loss throughout this work, however, our presentation holds for any general convex loss function.

### 4.2    NEWTON'S METHOD & THE GAUSS-NEWTON MATRIX

Full second-order optimization requires full access to the Hessian $H$, which can be used to precondition the gradient in each parameter update. This is known as Newton's method, and results in the following update rule:

$$\theta^* = \theta - H^{-1} g$$

In practice, for neural networks, the Hessian is not guaranteed to be positive semi-definite (PSD), and therefore Newton's method does not guarantee that the loss decreases in each iteration or even converges. As a consequence, it is common to instead use the *Gauss-Newton* matrix.

The Gauss-Newton matrix is defined to be the first term of the following decomposition of the Hessian, where $z := f(x)$ denotes the pre-softmax outputs of the model $f$ and $a$ goes over the output dimensions of $f$:

$$\nabla_\theta^2 \mathcal{L}(\theta) = \underbrace{\nabla_\theta f(\theta)^\top \nabla_z^2 \mathcal{L}(\theta) \nabla_\theta f(\theta)}_{\text{Gauss-Newton matrix}} + \sum_a \frac{\delta \mathcal{L}}{\delta z_a} \nabla_\theta^2 [f(\theta)]_a$$

That is, the Gauss-Newton matrix is defined as $G := \nabla_\theta f(\theta)^\top \nabla_z^2 \mathcal{L}(\theta) \nabla_\theta f(\theta)$. Intuitively, the Gauss-Newton captures the curvature of the loss function, but drops the curvature of the model.

---

[1]We refer to diagonal preconditioners as first-order and non-diagonal as second-order.

Unlike the Hessian, the Gauss-Newton matrix is PSD for MSE and cross-entropy loss (Martens, 2020). This avoids untrustworthy updates as negative curvature implies unbounded decrease in loss (Martens, 2020). Indeed, methods using the Gauss-Newton matrix rather than the full Hessian have been found to lead to better optimization (Martens, 2010; Martens & Sutskever, 2012; Vinyals & Povey, 2012).

### 4.3 MEMORY-FEASIBLE GAUSS-NEWTON IMPLEMENTATION

To test the limits of second-order optimizers, we want to apply the full Gauss-Newton term as the preconditioner. Formally, for gradient $g$, Gauss-Newton matrix $G$ and current parameters $\theta$, the Gauss-Newton update is

$$\theta^* = \theta - G^{-1}g \tag{1}$$

However, given that computing the Gauss-Newton matrix directly is infeasible, we instead run a functionally equivalent method that leverages Jacobian-vector products (JVPs) to avoid explicitly storing the Hessian. Specifically, we optimize the second-order Taylor approximation of the loss function $\mathcal{L}$ with a first-order Taylor approximation of the model $f$. The minimization of the loss in this setting is equivalent to using the Gauss-Newton matrix as a preconditioner (Martens & Sutskever, 2011b). The proof is provided in Appendix A.

We are now ready to define our Gauss-Newton method. Let $\widetilde{\mathcal{L}}_{\theta_t}(\theta) := \mathcal{L}(f^{(1)}_{\theta_t}(\theta, x), y)$ be the loss function on the first-order Taylor expansion of $f$ around current parameters $\theta_t$. Let $\widetilde{\mathcal{L}}^{(2)}_{\theta_t}(\theta)$ denote the second-order Taylor expansion of $\widetilde{\mathcal{L}}$ around $\theta_t$.

Given current parameters $\theta_t$, we define the Gauss-Newton update (Algorithm 1) as

$$\theta^* = \operatorname{argmin}_\theta \widetilde{\mathcal{L}}^{(2)}_{\theta_t}(\theta)$$

With this definition, there remains the problem of finding the minimizing $\theta^*$. As it is difficult to solve for the minimum directly, we instead use a separate optimizer to minimize $\widetilde{\mathcal{L}}^{(2)}_{\theta_t}(\theta)$. In our experiments we use Muon (Jordan et al., 2024b) as this "inner optimizer" as we found it to outperform AdamW. More details on this inner optimization procedure are given in Section 5.

For the GN-prox-linear algorithm (Algorithm 2), we instead define the updated iterate as

$$\theta^* = \operatorname{argmin}_\theta \widetilde{\mathcal{L}}_{\theta_t}(\theta).$$

The results for GN-prox-linear algorithm are provided in Section 6.2.

## 5 EXPERIMENT DETAILS

**Training details** We train 45M and 150M parameter LLaMA models (Touvron et al., 2023) on the C4 dataset (Raffel et al., 2020). Full details on models and hyperparameter sweeps are given in Appendix E and Appendix H respectively.

**Baselines** We run AdamW (Loshchilov & Hutter, 2019), Muon (Jordan et al., 2024b), and SOAP (Vyas et al., 2025) as baselines. For 45M models, for each method at each batch size, we run a hyperparameter sweep over learning rate, weight decay, and weight averaging decay if applicable. We additionally sweep the $\beta_2$ parameter for Adam, the $\mu$ parameter for Muon, and $(\beta_1, \beta_2)$ for SOAP. For 150M parameter models we run a more limited hyperparameter sweep over learning rate and $\beta$ and $\mu$ parameters. To make sure runs are well-initialized, we start all runs after an AdamW warmup consisting of $5\%$ of the Chinchilla-optimal number of tokens (Hoffmann et al., 2022). More details on hyperparameter tuning are given in Appendix H.

**Gauss-Newton** For each training step, we take a first-order Taylor approximation of the model around the current parameters. We initialize the parameters of the Taylor approximation to be the pre-linesearch parameters from the previous iteration (see Section 5.1). We then take a second-order Taylor approximation around the cross-entropy loss on the Taylorized model, also around the current model parameters. We use Muon (Jordan et al., 2024b) with batch size $b_{inner}$ to minimize

the Taylorized loss. We take $N$ steps of Muon and then update the model parameters using a line search. We refer to the global batch size (in number of sequences) as $b = N \times b_{inner}$ since this is the total amount of data seen per parameter update on the true model. We use $b_{inner} = 32$ for the 45M models and $b_{inner} = 128$ for the 150M models with sequence length 1024, and vary $N$ to control the overall batch size $b$. We start all runs from the same AdamW post-warmup checkpoint. To compute the necessary Taylor approximations we use the `neural-tangents` library from Novak et al. (2020). See Algorithm 1 for details. In addition we provide an efficient implementation for the minimization of the least squares subproblem in Appendix B.

**Upper bound for Gauss-Newton method**   In our experiments, we use Muon (Jordan et al., 2024b) with batch size $b_{inner}$ to solve the least squares problem in Algorithm 1. Therefore, Muon with batch size $b_{inner}$ trained on the true model and loss marks the upper bound for the Gauss-Newton method: since Muon in our method optimizes over the respective Taylor approximations, it is upper bounded by the performance of Muon with the same batch size on the true model and loss. We include results for Muon with batch size $b_{inner}$ in order to judge the relative performance of the Gauss-Newton method.

**GN-prox-linear**   For each training step, we minimize the loss of the first order Taylor expansion of the model around the current parameters (as mentioned in Algorithm 2). This method evaluates whether incorporating higher-order terms in the loss function yields improvements compared to the Gauss–Newton method. The results for this method are discussed in Section 6.2.

## 5.1   OPTIMIZATION STRATEGIES

We perform extensive hyperparameter sweeps, learning rate scheduling strategies, and regularization strategies to test the limits of the Gauss-Newton method.

**Learning rate schedules**   We experiment with three learning rate schedules for Gauss-Newton runs, which we refer to as "global cosine," "global+inner cosine," and "constant+inner cosine." We depict each learning rate schedule in Figure 4.

**Regularization**   We experiment with several types of regularization strategies to improve the stability of the Gauss-Newton runs at high learning rate. These fall into two categories of *inner optimization* and *outer optimization* regularization strategies. For inner optimization strategies (regularization involving the inner optimization loop to solve the least squares problem), we add weight decay to the optimizer as well as a weight decay term to the loss, which adds regularization on the $\ell_2$ norm of the magnitude of the parameter update. For outer optimization, we experiment with line search to control the size of the parameter update.

**Inner optimizer**   We note that the choice of inner optimizer has a significant impact on the upper bound of the Gauss-Newton method; we consider our results with respect to the performance of the inner optimizer. Regardless, we find that Muon outperforms AdamW as the inner optimizer for the Gauss-Newton runs.

**Takeaways from optimization strategies**   We found that **learning rate schedule** and **line search** had major impact on the stability of training for the Gauss-Newton method. As for learning rate schedules, we find that the global cosine schedule outperforms the global+inner cosine schedule at small to medium batch size, but the constant+inner cosine schedule can be helpful for runs at large batch size for Gauss-Newton. Additionally, we found line search to be essential for stable convergence for Gauss-Newton runs. However, we found that when using line search, it helps to set the initial parameters for the next inner minimization to be the *pre-linesearch* parameters from the previous step. These findings coincide with those of Martens (2010), which finds that sharing information across iterations and backtracking improve performance of a conjugate gradient-based Hessian-free optimization strategy. The importance of inner optimizer and sharing information across iterations seem to imply that we are not finding the precise Gauss-Newton update at each step – it is possible that with further optimization the Gauss-Newton method could achieve even better performance.

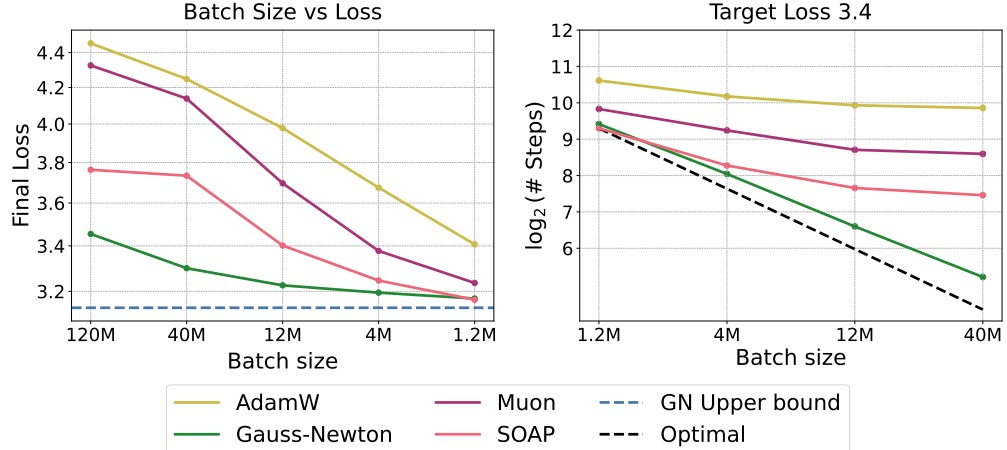

Figure 2: **Left:** Batch size vs final validation loss for models trained for Chinchilla-optimal number of tokens. The dotted line marks the loss achieved by a model trained with Muon with batch size 128k. This represents the upper bound of performance for our Gauss-Newton method. **Right:** Critical batch size scaling. The dotted line marks the optimal scaling trend, where no sample efficiency is lost as batch size increases.

# 6 EXPERIMENTS

## 6.1 GAUSS-NEWTON EXPERIMENTS

### 6.1.1 ITERATION COMPLEXITY

We measure the iteration complexity of each method by measuring the number of steps it takes to reach loss 3.25 with extremely large batch size. Specifically, for each method, we use a batch size significantly beyond that method's critical batch size (McCandlish et al., 2018; Shallue et al., 2019) such that further increasing batch size does not reduce the number of training steps needed to achieve a given performance.[2] Following our critical batch size findings in Section 6.1.2, we use a batch size of 40M tokens for AdamW and Muon as gains disappear almost completely beyond this amount, and a batch size of 240M tokens for SOAP and Gauss-Newton. We choose this threshold and these batch sizes to enter the regime in which additional batch size increases no longer reduce the required steps, while keeping runs feasible.

We find that the Gauss-Newton method can make fast progress in the large batch size regime, particularly in the first few steps of training. After 10 steps, the loss for the Gauss-Newton model is below 3.75, while other methods have made marginal progress from the starting loss. The optimal learning rates for AdamW, Muon, and SOAP lead to initial instability but faster convergence overall – see Appendix H for details on choosing hyperparameters. The Gauss-Newton method is able to reach loss 3.25 in 54 steps, a 5.4x gain over SOAP and 16x gain over Muon. The results are shown in Figure 1.

### 6.1.2 BATCH SIZE SCALING

While iteration complexity captures a purely sequential perspective of training efficiency, it is also important to consider the sample efficiency. In an ideal setting, the number of samples seen at each step would vary proportionally to the number of steps, such that there is always a constant number of samples required overall. However, it is known that sample efficiency is lost once the batch size is scaled past a given method's *critical batch size* (McCandlish et al., 2018; Shallue et al., 2019). That is, the total number of samples needed to achieve a given loss will grow once the critical batch size is exceeded. Therefore, we study the batch size scaling behavior of the Gauss-Newton

---

[2]Since batch sizes are increased using gradient accumulation in our experiments, we choose batch size based on each method's critical batch size to save compute.

method to understand how much we lose in sample and computational efficiency when we minimize the number of sequential iterations. We run experiments in two settings: first, we train models for a fixed amount of data over a range of batch sizes and measure the final loss. Second, we measure how the number of steps to achieve a given loss changes with increasing batch size. We run experiments for 45M- and 150M-parameter models; see Appendix G for results at 45M-parameter scale.

**Training for fixed token count**  We train 150M-parameter models for 3B tokens following Chinchilla-optimal scaling laws (Hoffmann et al., 2022), ranging batch size from 1.2M to 120M tokens. We observe similar performance between SOAP and Gauss-Newton up to batch size 4M, while substantial gains are achieved by Gauss-Newton for larger batch size (Figure 2). Of existing methods, SOAP performs best, followed by Muon and then AdamW. Especially noteworthy is the performance of the Gauss-Newton method at batch size 120M, which uses only 20 steps of optimization. Here we are able to achieve loss 3.45 with Gauss-Newton. For comparison, AdamW achieves loss 3.4 with batch size 1.2M, and degrades to loss above 4.4 with batch size 120M.

**Training to reach a target validation loss**  Following the methodology of Zhang et al. (2025), we plot the number of steps required for each optimization method to reach the target validation loss of 3.4 as a function of batch size. The point at which the curve for each model plateaus defines its respective *critical batch size* (McCandlish et al., 2018; Shallue et al., 2019; Jain et al., 2018b). We find that AdamW levels off near batch size of 4M with little further reduction. SOAP and Muon continue to decrease up to batch size of 12M but with diminishing reductions, and show little additional decrease by 40M. Meanwhile, the Gauss-Newton method continues to decrease through 40M, indicating better sample efficiency at large batch sizes.

## 6.2 GN-PROX-LINEAR METHOD

We define a variation of our method that corresponds to another convex problem that retains the full loss function on the linearized model instead of using the second-order approximation. This method follows the same procedure as the Gauss-Newton method in Section 4 but directly minimizes the loss on the linearized model, denoted by $\widetilde{\mathcal{L}}_{\theta_t}(\theta)$:

$$\theta^* = \mathrm{argmin}_\theta \ \widetilde{\mathcal{L}}_{\theta_t}(\theta)$$

See Algorithm 2 for details. Note that for any convex loss function (including cross-entropy loss), the above optimization problem is still convex. Moreover, this problem is related to the richly studied literature of kernelized classification (Shalev-Shwartz & Ben-David, 2014) (albeit with cross-entropy loss, instead of the max-margin loss). This GN-prox-linear method is notably not a second-order method. Rather, it allows us to study the effect of the higher order terms of the loss as compared to the Gauss-Newton update rule.

We train in the same Chinchilla-optimal setting on 150M parameter models and perform the same hyperparameter sweeps as for Gauss-Newton (See Appendix H). We find that the inclusion of higher order loss terms has little effect on performance as compared to Gauss-Newton; results are shown in Figure 3. However, unlike Gauss-Newton, we found that the global cosine schedule for the inner optimizer outperformed the constant+inner cosine schedule. Additionally, line search was not necessary for the GN-prox-linear method.

## 6.3 LAYERWISE GAUSS-NEWTON

Many existing second-order optimizers use layerwise approximations of the Hessian for computational and memory feasibility. This prompts a further study on whether the full Hessian is necessary to achieve the performance gains discussed in Section 5. Specifically, we want to understand the importance of the cross-layer Hessian information.

We define a layerwise version of our Gauss-Newton method, in which we take a Taylor expansion around each model layer and optimize the second-order Taylor expansion of the loss separately for each layer.

Formally, let $\theta_{l,t}$ be the set of parameters at time $t$ for layer $l$ of the network. For layer $l$, define $f_{\theta_{l,t}}^{(1)}(\theta_l)$ as the first-order Taylor expansion of $f$ with respect to only the parameters $\theta_l$, expanded

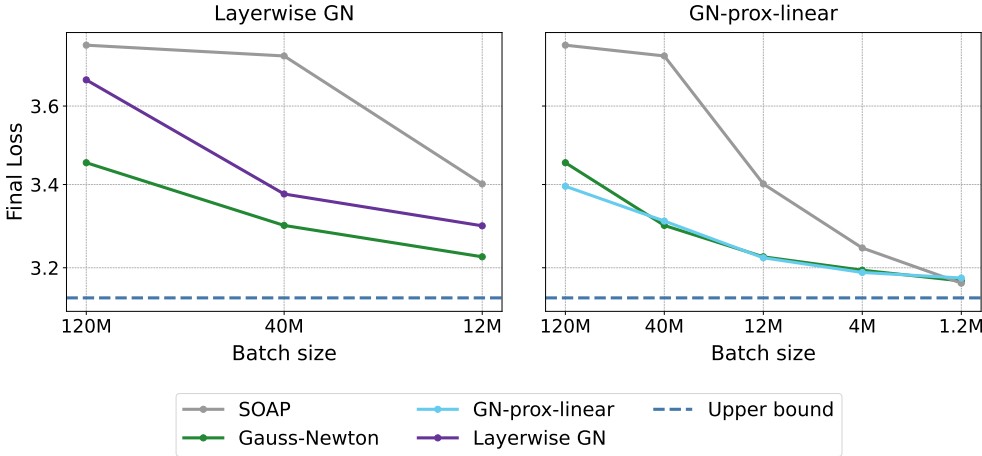

Figure 3: **Left**: Comparison of Gauss-Newton to the layerwise implementation for Chinchilla-optimal token count for 150M parameter models. The layerwise method achieves almost matching performance to that of the full Gauss-Newton. **Right**: The Gauss-Newton update closely matches the GN-prox-linear method that has access to higher order loss terms.

around the current parameters $\theta_{l,t}$, while keeping the parameters of all other layers fixed at their current values.

At each timestep $t$, take $\widetilde{\mathcal{L}}^{(2)}_{\theta_{l,t}}(\theta_l)$ to be the second-order Taylor expansion of the loss on $f^{(1)}_{\theta_{l,t}}(\theta_l)$. Then for each layer, we solve for

$$\theta_{l,t+1} = \mathrm{argmin}_{\theta_l}\, \widetilde{\mathcal{L}}^{(2)}_{\theta_{l,t}}(\theta_l)$$

After independently computing updates for each layer, we merge the updated layer parameters. We then apply a line search over the merged parameter set to obtain the final parameter update at step $t$.

Due to compute costs, we train 150M parameter layerwise Gauss-Newton models only for the largest three batch sizes. We follow the same training setting for fixed token count as specified in Section 6.1.2. For layerwise experiments we set hyperparameters to match those of the best Gauss-Newton configuration at each batch size. However, we include smaller step size options for the line search as we find this is necessary for stable convergence. We provide more details on line search step sizes in Appendix D. We find that the layerwise Gauss-Newton method also achieves comparable performance through batch size of 40M tokens (Figure 3). We additionally train a layerwise Gauss-Newton model with batch size of 120M tokens to loss 3.25 to compare its iteration complexity to that of full Gauss-Newton (See Sec 6.1.1). We find that the layerwise Gauss-Newton takes only 1.4x more steps to reach the target loss compared to the full Gauss-Newton method and provides a 3.4x gain over SOAP (Figure 1).

## 7 DISCUSSION AND CONCLUSION

In this work, we study whether full second-order optimization – specifically, using the full Gauss-Newton matrix as a preconditioner – can offer further benefits for training large language models as compared to existing methods. In particular, we focus on the large batch size regime, following Jain et al. (2018a) and Zhang et al. (2019) which show that the benefits of preconditioning may not appear at small batch size. While our current implementation is roughly 1.5x slower than standard training (e.g. with AdamW or Muon), we view this as a proof of concept demonstrating the potential of exact second-order methods. Our results indicate that further development in second-order methods could lead to substantial benefits in convergence and ability to scale to larger batch size.

While we perform extensive hyperparameter sweeps and regularization strategies, we acknowledge that there could be other optimization strategies to further improve the performance of the Gauss-Newton method. In addition, our work is limited to applying the inverse of the Gauss-Newton matrix

as the preconditioner ($G^{-1}$). There may be better ways to apply full second-order optimization for large language models. We encourage future work in this area and hope our findings are informative.

We also compare Gauss-Newton to the GN-prox-linear method to study whether there is benefit to including higher order loss terms beyond second-order. Our results suggest that Gauss-Newton can achieve performance similar to this method, indicating that higher-order loss terms are not necessary to achieve gains in performance over current methods. In addition, our layerwise Gauss-Newton experiments suggest that better approximations to the per-layer Hessian may be sufficient to achieve substantial performance benefits over current methods. We encourage future work in developing computationally efficient and practical optimization methods in this direction.

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

## A    PROOF OF EQUIVALENCE TO GAUSS-NEWTON METHOD

In our method, we compute the Gauss-Newton update by minimizing the second-order Taylor approximation of the loss around the first-order Taylor approximation of the function.

Taking the first-order Taylor approximation of $f$ around $\theta_t$,

$$f_{\theta_t}^{(1)}(\theta, x) := f(\theta_t, x) + \nabla f(\theta_t, x)^\mathsf{T}(\theta - \theta_t)$$

Let $\widetilde{\mathcal{L}}_{\theta_t}(\theta) := \mathcal{L}(f_{\theta_t}^{(1)}(\theta, x), y)$ be the loss function on the Taylor expansion of $f$ around $\theta_t$.

Then the second-order Taylor approximation of $\widetilde{\mathcal{L}}_{\theta_t}(\theta)$ around $\theta_t$ gives

$$\widetilde{\mathcal{L}}_{\theta_t}^{(2)}(\theta) := \widetilde{\mathcal{L}}_{\theta_t}(\theta_t) + \nabla\widetilde{\mathcal{L}}_{\theta_t}(\theta_t)^\mathsf{T}(\theta - \theta_t) + \frac{1}{2}(\theta - \theta_t)^\mathsf{T}\nabla^2\widetilde{\mathcal{L}}_{\theta_t}(\theta_t)(\theta - \theta_t)$$

Denoting $z = f_{\theta_t}^{(1)}(\theta, x)$ and applying the chain rule for the gradients, we have

$$\nabla\widetilde{\mathcal{L}}_{\theta_t}(\theta) = \mathcal{L}'(f_{\theta_t}^{(1)}(\theta, x), y)\nabla f_{\theta_t}^{(1)}(\theta, x)$$

$$\nabla^2\widetilde{\mathcal{L}}_{\theta_t}(\theta) = \nabla f_{\theta_t}^{(1)}(\theta, x)^\top \nabla_z^2\mathcal{L}(f_{\theta_t}^{(1)}(\theta, x), y)\nabla f_{\theta_t}^{(1)}(\theta, x)$$

Then substituting and evaluating at $\theta_t$:

$$\nabla\widetilde{\mathcal{L}}_{\theta_t}(\theta)|_{\theta=\theta_t} = \mathcal{L}'(f(\theta_t, x), y)\nabla f(\theta_t, x)$$

$$\nabla^2\widetilde{\mathcal{L}}_{\theta_t}(\theta)|_{\theta=\theta_t} = \nabla_\theta f(\theta_t, x)^\top \nabla_z^2\mathcal{L}(f(\theta_t, x), y)\nabla_\theta f(\theta_t, x)$$

Then

$$\widetilde{\mathcal{L}}_{\theta_t}^{(2)}(\theta) = \mathcal{L}(f(\theta_t, x), y) + \mathcal{L}'(f(\theta_t, x), y)\nabla f(\theta_t, x)(\theta - \theta_t)$$
$$+ \frac{1}{2}(\theta - \theta_t)^\mathsf{T}\nabla_\theta f(\theta_t, x)^\top \nabla_z^2\mathcal{L}(f(\theta_t, x), y)\nabla_\theta f(\theta_t, x)(\theta - \theta_t)$$

Let $g$ denote the gradient of the loss at $\theta_t$, i.e. $\mathcal{L}'(f(\theta_t, x), y)\nabla_\theta f(\theta_t, x)$. Let G denote the Gauss-Newton term $\nabla_\theta f(\theta_t, x)^\top \nabla_z^2\mathcal{L}(f(\theta_t, x), y)\nabla_\theta f(\theta_t, x)$. Then we can write

$$\widetilde{\mathcal{L}}_{\theta_t}^{(2)}(\theta) = \mathcal{L}(f(\theta_t, x), y) + g(\theta - \theta_t) + \frac{1}{2}(\theta - \theta_t)^\mathsf{T}G(\theta - \theta_t)$$

Since the Gauss-Newton matrix is PSD, we can find $\theta^*$ to minimize $\widetilde{\mathcal{L}}_{\theta_t}^{(2)}$ by setting its gradient to zero:

$$g + (\theta^* - \theta_0)G = 0$$

which results in the update rule

$$\theta^* = \theta_0 - G^{-1}g$$

# B  IMPLEMENTATION OF GAUSS-NEWTON

Following notation from Appendix A, we have the Gauss-Newton objective:

$$\widetilde{\mathcal{L}}_{\theta_t}^{(2)}(\theta) = \mathcal{L}(f(\theta_t, x), y) + \mathcal{L}'(f(\theta_t, x), y)^\mathsf{T} \nabla f(\theta_t, x)(\theta - \theta_t)$$
$$+ \frac{1}{2}(\theta - \theta_t)^\mathsf{T} \nabla_\theta f(\theta_t, x)^\top \nabla_z^2 \mathcal{L}(f(\theta_t, x), y) \nabla_\theta f(\theta_t, x)(\theta - \theta_t)$$

Then the gradient is:

$$\nabla_\theta \widetilde{\mathcal{L}}_{\theta_t}^{(2)}(\theta) = \nabla f(\theta_t, x)^\mathsf{T} \mathcal{L}'(f(\theta_t, x), y) + \nabla_\theta f(\theta_t, x)^\top \nabla_z^2 \mathcal{L}(f(\theta_t, x), y) \nabla_\theta f(\theta_t, x)(\theta - \theta_t)$$

We compute the gradient directly in the following manner, which results in only one additional forward pass as compared to a standard backpropagation step:

---

**Algorithm 3:** Gauss-Newton Inner Optimization Step

---

**Input:** train_state $(\theta)$,
reference params $\theta_0$,
batch $\mathcal{B} = \{\text{input}, \text{target}\}$

**Define** $f_{\text{batch}}(\theta)$:
    logits $= f(\theta, \mathcal{B}.\text{input})$
    **return** logits                          `// f on current batch B`

**Define** $\ell_{\text{logits}}(y)$:
    loss $\leftarrow$ CE_loss$(y, \mathcal{B}.\text{target})$
    **return** loss

**Define** VALUEANDGRADIENT$(\theta_0, \theta)$:
    `// Linearize f at` $\theta_0$`:` $f(\theta) \approx y_0 + J_0(\theta - \theta_0)$
    $(y_0, \text{jvp}) \leftarrow$ linearize$(f_{\text{batch}}, \theta_0)$
    $d\theta \leftarrow \theta - \theta_0$
    $v \leftarrow$ jvp$(d\theta)$                                  `//` $v = J_0 \, d\theta$

    `// Logit-space gradient and Hessian-vector`
    $\nabla_y \ell_{\text{logits}} \leftarrow$ grad$(\ell_{\text{logits}})$
    $(g_0, H_y v) \leftarrow$ JVP$(\nabla_y \ell_{\text{logits}}, (y_0), (v))$         `//` $H_y = \nabla_{yy}^2 \ell$ `at` $y_0$

    `// Pullback through the linearization:` $J_0^\top (g_0 + H_y v)$
    jt $\leftarrow$ linear_transpose$(\text{jvp}, \theta_0)$
    $\nabla_\theta \leftarrow$ jt$(g_0 + H_y v)$

    `// Quadratic objective on the linear model`
    loss $\leftarrow \ell_{\text{logits}}(y_0) + g_0 \cdot v + \frac{1}{2} v \cdot (H_y v)$
    **return** $(\text{loss}, \nabla_\theta)$

loss, $g \leftarrow$ VALUEANDGRADIENT$(\theta_0, \theta = \text{train\_state.params})$

**Apply update:**
train_state $\leftarrow$ train_state.apply_gradients$(g)$

**return** train_state

---

## C  ADDITIONAL RELATED WORK

Also related to our work are optimizers that leverage diagonal approximations to the Hessian, such as AdaHessian (Yao et al., 2021) and Sophia (Liu et al., 2024). These propose lightweight approximations to the diagonal Hessian rather than layerwise approximations as in Shampoo and SOAP. However, Zhao et al. (2025) show that Sophia performs comparably to AdamW, suggesting the need to go beyond the diagonal Hessian.

## D  ADDITIONAL DETAILS ON OPTIMIZATION STRATEGIES

### D.1  LEARNING RATE SCHEDULES

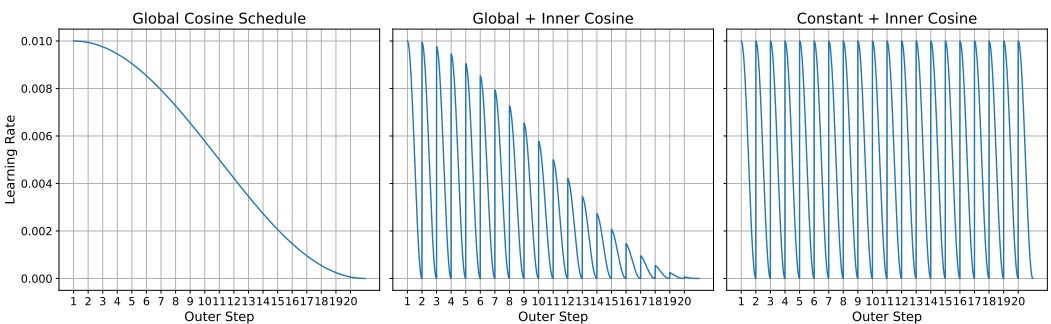

Figure 4: Three learning rate schedules used for the Gauss-Newton and GN-prox-linear runs. From left to right: global cosine, global+inner cosine, and constant+inner cosine. Each inner cosine period lasts the duration of the optimization over the current Taylor expansion; outer step refers to each parameter update on the model.

We experiment with three learning rate schedules: "global cosine," "global+inner cosine," and "constant+inner cosine." From preliminary experiments we find that global+inner cosine did not generally outperform the global cosine and constant+inner cosine options, so we do not use global+inner cosine in our main experiments.

### D.2  LINE SEARCH STEP SIZES

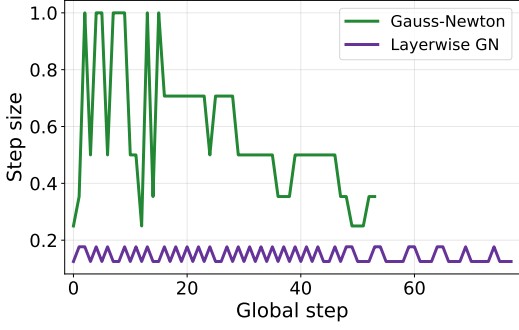

Figure 5: Resulting step sizes used from line search for Gauss-Newton and layerwise Gauss-Newton.

We find that line search is necessary for the performance of Gauss-Newton and layerwise Gauss-Newton runs. For the line search, we evaluate the true model loss on step sizes $2^{\frac{-i}{2}}$ with $i \in \{0, \ldots, 4\}$ for full Gauss-Newton and $i \in \{0, \ldots, 9\}$ for layerwise Gauss-Newton. Figure 5 shows the step sizes at each step for iteration complexity runs (See Section 6.1.1).

## E  MODEL DETAILS

| Configuration | 45M Model | 150M Model |
|---|---|---|
| Hidden Size | 512 | 768 |
| Intermediate Size | 2048 | 3072 |
| Number of Layers | 4 | 12 |
| Attention Heads | 8 | 16 |
| Key/Value Heads | 8 | 16 |

Table 1: Model configurations for the 45M and 150M parameter LLaMA-based models.

## F Compute resources

All 45M parameter model runs are trained on 1 Nvidia 80GB H100. 45M runs for AdamW, Muon, and SOAP trained for 2-3 hours and Gauss-Newton and GN-prox-linear trained for 1-2 days. 150M parameter model runs for batch size scaling experiments for AdamW, Muon, and SOAP are each trained using 1 H100 for roughly 6-20 hours. 150M parameter model runs for batch size scaling experiments for Gauss-Newton and GN-prox-linear are each trained for 1-3 days with 4 80GB Nvidia H100s using distributed data parallel (DDP). Layerwise Gauss-Newton runs each trained for 3-7 days with 4 H100s. Iteration complexity runs trained for 2-3 days for AdamW and Muon and 15-30 days for SOAP, Gauss-Newton, and Layerwise Gauss-Newton.

## G Experiments on 45M parameter models

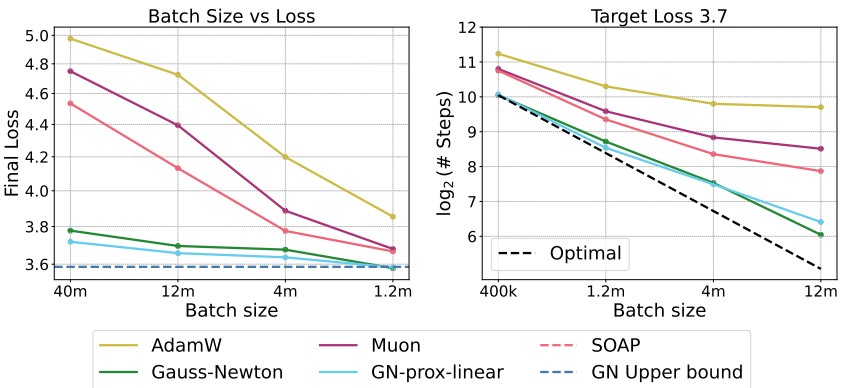

Figure 6: **Left:** Batch size vs final validation loss for 45M parameter models. All models are trained for 900M tokens on the C4 dataset (Raffel et al., 2020) following Chinchilla scaling laws (Hoffmann et al., 2022). The dotted line marks the loss achieved by a model trained with Muon with a batch size of 32k tokens. This represents the upper bound of performance for our Gauss-Newton and GN-prox-linear methods as we use Muon with batch size of 32k tokens as the inner optimizer to compute the parameter update in each step. **Right:** Critical batch size scaling for 45M parameter models. The dotted line marks the optimal scaling trend, where no sample efficiency is lost as batch size increases.

## H  DETAILS ON HYPERPARAMETER TUNING

| Shared Hyperparameters | |
|---|---|
| Model Size | 45M |
| Batch Size | 400k, 1.2m, 4m, 12m, 40m |
| Context Length | 1024 |
| **AdamW** | |
| Learning Rate | 0.001, 0.003, 0.01, 0.03 |
| Weight Decay | 0, (0.001, 0.01) |
| Additional Warmup Fraction | 0, (0.1) |
| Momentum $\beta_1$ | 0.9 |
| Adam $\beta_2$ | 0.95, 0.99, 0.999 |
| LR Scheduler | constant+EWA, cosine |
| EWA Decay Rate $\tau$ | 0.9, 0.99 |
| **Muon** | |
| Learning Rate | 0.03, 0.1, (0.3) |
| Additional Warmup Fraction | 0 |
| Momentum $\mu$ | 0.9, 0.95, 0.99 |
| LR Scheduler | constant+EWA, cosine |
| EWA Decay Rate $\tau$ | (0.3), 0.5, 0.7, (0.9) |
| **SOAP** | |
| Preconditioning Frequency | 1 |
| Learning Rate | 0.01, 0.03, (0.1) |
| Additional Warmup Fraction | 0 |
| Momentum $\beta_1$ | 0.7, 0.8, 0.9 |
| Adam $\beta_2$ | 0.7, 0.8, 0.9 |
| LR Scheduler | constant+EWA, cosine |
| EWA Decay Rate $\tau$ | (0.5), 0.7, 0.8, (0.9) |
| **Gauss-Newton** | |
| Learning Rate | (0.001), 0.003, 0.01, 0.03, (0.1) |
| Inner Loop Warmup Fraction | 0, (0.2) |
| Global Warmup Fraction | 0, (0.2) |
| Momentum $\mu$ | 0.95 |
| Optimizer weight decay | (0), 0.001 |
| Loss weight decay | 0, 0.01, (0.1) |
| Parameter weight decay | 0, (0.01, 0.03, 0.1) |
| LR Scheduler | constant+inner cosine, global cosine |
| Linesearch | True, False |
| EWA Decay Rate $\tau$ | (0.99), 0.999 |

Table 2: Hyperparameter configurations used for 45M models. Values in parentheses were not used for every sweep. For the critical batch size plot (Figure 6 right) only the constant+EWA learning rate schedule was used. We start each run after an AdamW warmup of 5% for 45M parameter models; additional warmup refers to warmup starting from this checkpoint. For baselines, weight averaging is used only with the constant schedule. All Gauss-Newton runs without line search use weight averaging; runs with line search use no weight averaging. Inner loop warmup applies only to the constant+inner cosine schedule.

| Shared Hyperparameters | |
|---|---|
| Model Size | 150M |
| Batch Size | 1.2m, 4m, 12m, 40m, 120m |
| Context Length | 1024 |
| **AdamW** | |
| Learning Rate | 0.001, 0.003, 0.01 |
| Weight Decay | 0.001, 0.1 |
| Additional Warmup Fraction | 0, (0.1) |
| Momentum $\beta_1$ | 0.9 |
| Adam $\beta_2$ | 0.95, 0.99 |
| LR Scheduler | cosine |
| **Muon** | |
| Learning Rate | 0.01, 0.03, 0.1 |
| Weight Decay | 0 |
| Additional Warmup Fraction | 0 |
| Momentum $\mu$ | 0.9, 0.95, 0.99 |
| LR Scheduler | cosine, (constant+EWA) |
| EWA Decay Rate $\tau$ | 0.7, 0.9 |
| **SOAP** | |
| Preconditioning Frequency | 1 |
| Learning Rate | 0.01, 0.03, 0.1 |
| Weight Decay | 0 |
| Additional Warmup Fraction | 0 |
| Momentum $\beta_1$ | (0.7), 0.9, (0.95) |
| Adam $\beta_2$ | 0.7, 0.9, 0.95 |
| LR Scheduler | cosine, (constant+EWA) |
| EWA Decay Rate $\tau$ | 0.7, 0.9 |
| **Gauss-Newton** | |
| Inner Loop Learning Rate | 0.003, 0.01, 0.03, 0.1 |
| Inner Loop Warmup Fraction | 0, 0.2 |
| Global Warmup Fraction | 0 |
| Inner Loop Weight Decay | 0, 0.01, 0.1 |
| Optimizer Weight Decay | 0.001 |
| Momentum $\mu$ | 0.95 |
| LR Scheduler | constant+inner cosine, global cosine |
| Linesearch | True, False |
| EWA Decay Rate $\tau$ | (0.9, 0.99), 0.999 |

Table 3: Hyperparameter configurations used for 150M models for batch size scaling experiments (Section 6.1.2). For Muon, constant+EWA learning rate schedule was included in the sweep for the three largest batch sizes. For Gauss-Newton, due to high compute costs we hand-tune over the range of provided values rather than conducting the entire sweep at each batch size. Inner loop warmup applies only to the constant+inner cosine schedule. We start each run after an AdamW warmup of 5% for 150M parameter models; additional warmup refers to warmup starting from this checkpoint. For baselines, weight averaging is used only with the constant schedule. All Gauss-Newton runs without line search use weight averaging with $\tau = 0.999$; runs with line search are default to no weight averaging or are swept with lower values of $\tau$. For iteration complexity experiments (Section 6.1.1): For AdamW and Muon we take the best hyperparameters from sweeps at 40M batch size in the batch size experiments. For SOAP and Gauss-Newton we use the best hyperparameters from sweeps at 120M batch size and run a limited sweep over learning rate.

