# OpenReview forum: "The Potential of Second-Order Optimization for LLMs: A Study with Full Gauss-Newton"
_ICLR.cc/2026/Conference — ICLR 2026 Poster_

### Official Review · Reviewer_Td6r · 2025-10-30

**Soundness:** 4
**Presentation:** 4
**Contribution:** 3
**Rating:** 4
**Confidence:** 5

**Summary:**

The authors of this manuscript investigate the practical upper bound of second-order optimization methods for training Large Language Models (LLMs). They establish this bound by applying a full Gauss-Newton (GN) preconditioner to transformer models of up to 150M parameters, comparing its iteration complexity against strong baselines like SOAP and Muon. They report that the full GN update achieves a significant reduction in training iterations (e.g., 5.4x over SOAP). Furthermore, they find that a precise layerwise GN preconditioner, which ignores cross-layer information, achieves nearly identical performance to the full GN method.

**Strengths:**

1- Studies showing the capability and limitation of current and ideal second-order optimization methods are very useful and insightful.

2- The paper is well-written and addresses a clear and important question: what is the practical performance ceiling for second-order methods in LLM training?

3- The finding that a precise layerwise Gauss-Newton preconditioner nearly matches the full GN method is a very strong and useful insight, as it suggests where future efforts in developing practical algorithms should be focused.

**Weaknesses:**

1- The current study lacks a comparison to an important family of second-order optimizers: the KFAC [1] family, which is orthogonal to the Shampoo-family methods (SOAP, Muon) that were analyzed. KFAC optimizers have been successfully scaled to large models, such as in KAISA [2] and MKOR [3]. It is important to compare the capabilities of such methods and understand how they relate to the Gauss-Newton "oracle" presented in this work.

2- In line 117, the authors claim that the most common optimizer used for LLMs is Adam, while AdamW [4] is more commonly used and is the default for many libraries (e.g., HuggingFace Trainer). While the authors use AdamW as a baseline, a direct comparison including both Adam and AdamW as baselines would be insightful to clarify the impact of decoupled weight decay relative to the second-order gains.

4- This study mainly focuses on the beginning part of the training process, where the loss decreases very rapidly (e.g., Figures 1 & 2 show convergence to a relatively high loss of 3.25-3.4). This is in contrast to the most dominant and expensive part of LLM pretraining, where the loss decrease is very gradual and almost plateaus. This focus is a limitation and leads to questioning the practicality of the findings for the full, end-to-end training process. I understand that scaling these optimizers (especially the full GN method) is very expensive, but without experiments that explore this later-stage convergence, the findings on iteration complexity gains remain inconclusive.

---

[1] Martens, J., & Grosse, R. (2015). Optimizing Neural Networks with Kronecker-factored Approximate Curvature. arXiv:1503.05671.

[2] Pauloski, J. G., et al. (2021). KAISA: An Adaptive Second-Order Optimizer Framework for Deep Neural Networks. SC21.

[3] Mozaffari, M., et al. (2023). MKOR: Momentum-Enabled Kronecker-Factor-Based Optimizer Using Rank-1 Updates. NeurIPS 2023.

[4] Loshchilov, I., & Hutter, F. (2019). Decoupled Weight Decay Regularization. ICLR 2019.

**Questions:**

1- How does the performance and scaling of the full Gauss-Newton oracle compare to the KFAC family of optimizers [1], including scaled implementations like KAISA [2] and MKOR [3]?

2- Could the authors provide an ablation that includes both Adam and the baseline AdamW [4] to clarify the impact of decoupled weight decay in this context?

3- How do the observed iteration complexity gains from GN hold up in the later stages of training, after the initial rapid loss decrease and into the long-tail plateau phase?

---

[1] Martens, J., & Grosse, R. (2015). Optimizing Neural Networks with Kronecker-factored Approximate Curvature. arXiv:1503.05671.

[2] Pauloski, J. G., et al. (2021). KAISA: An Adaptive Second-Order Optimizer Framework for Deep Neural Networks. SC21.

[3] Mozaffari, M., et al. (2023). MKOR: Momentum-Enabled Kronecker-Factor-Based Optimizer Using Rank-1 Updates. NeurIPS 2023.

[4] Loshchilov, I., & Hutter, F. (2019). Decoupled Weight Decay Regularization. ICLR 2019.

---

> ### Author Response · Authors · 2025-11-19
>
> We thank the reviewer for their time and thoughtful comments, and we provide responses below. We will first summarize two main changes in our updated version of the paper and then address the reviewer’s specific questions:
>
> (1) We have updated the presentation of Algorithm 2 (“Linearized model method”). We renamed this algorithm to “GN-prox-linear” since the algorithm essentially takes the form of a prox-linear method as in [1,2]. We have also updated the notation in the algorithm boxes for both Algorithm 1 and 2. We hope that this clarifies some confusion on the “Linearized model method” in particular.
>
> (2) We have implemented a more efficient method for the Gauss-Newton and GN-prox-linear methods. In our original experiments there was roughly 4x overhead for these methods compared to Muon, making experiments very expensive. The new implementation requires only one additional forward pass (+some negligible computation in the logits-space for Gauss-Newton), and results in roughly 1.5x overhead. We provide details for the new implementation in Appendix B as well as in our code. While our method is still not meant to be practical given that it still relies on an iterative Muon subroutine, the more efficient implementation enables us to more easily scale experiments.
>
> In addition to the updated paper, we provide an anonymous code repository here: https://anonymous.4open.science/r/full-gauss-newton-7528.
>
> To the reviewer’s specific concerns:
>
> > 1- How does the performance and scaling of the full Gauss-Newton oracle compare to the KFAC family of optimizers [1], including scaled implementations like KAISA [2] and MKOR [3]?
>
> Prior work suggests that KFAC performs comparably to Shampoo [3,4], so based on our findings comparing Gauss-Newton and SOAP we believe that Gauss-Newton would compare similarly to KFAC.
>
> > 2- Could the authors provide an ablation that includes both Adam and the baseline AdamW [4] to clarify the impact of decoupled weight decay in this context?
>
> Several previous works show that AdamW outperforms Adam [5,6], so we choose to focus on AdamW for our experiments. A comparison of decoupled weight decay is outside the scope of our work. We will double check that we refer to Adam vs AdamW correctly throughout the paper.
>
> > 3- How do the observed iteration complexity gains from GN hold up in the later stages of training, after the initial rapid loss decrease and into the long-tail plateau phase?
>
> Currently we have results extending the 3.25 loss target in Figure 1 to 3.2 and the trend continues to hold, but we will also launch some longer runs to investigate this. (Our new implementation should allow us to scale the number of steps beyond what was initially feasible).
>
> We thank you again for the helpful comments on our paper. We will report back when we have the new results.
>
> References:
>
> [1] James V. Burke. Descent methods for composite nondifferentiable optimization problems. Math. Program., 33(3): 260–279, December 1985. ISSN 0025-5610. doi: 10.1007/BF01584377. URL https://doi.org/10.1007/BF01584377.
>
> [2] Dmitriy Drusvyatskiy. The proximal point method revisited, 2017. URL https://arxiv.org/abs/1712.
> 06038.
>
> [3] Anil, Rohan, et al. "Scalable second order optimization for deep learning." arXiv preprint arXiv:2002.09018 (2020).
>
> [4] Ren, Yi, and Donald Goldfarb. "Tensor normal training for deep learning models." Advances in Neural Information Processing Systems 34 (2021): 26040-26052.
>
> [5] Loshchilov, Ilya, and Frank Hutter. "Decoupled weight decay regularization." arXiv preprint arXiv:1711.05101 (2017).
>
> [6] P. Zhou, X. Xie, Z. Lin and S. Yan, "Towards Understanding Convergence and Generalization of AdamW," in IEEE Transactions on Pattern Analysis and Machine Intelligence, vol. 46, no. 9, pp. 6486-6493, Sept. 2024, doi: 10.1109/TPAMI.2024.3382294.

---

> > ### Comment · Reviewer_Td6r · 2025-11-26
> >
> > I thank the authors for their clarifications, but most of my concerns are not addressed with justifiable results (specially the third concern requires actual data to validate the scope of this analysis).
> >
> > I would like to maintain my current score.

---

> ### Author Response · Authors · 2025-12-03
>
> To follow up on the reviewer's question about later stages in training: From continuing our previous iteration complexity runs for Gauss-Newton until loss 3.16 we end up with a 4.3x reduction of steps over SOAP. It seems like the GN subproblem requires careful tuning as loss continues to decrease. Unfortunately due to resource limitations during the rebuttal period we could not conduct a full sweep of hyperparameters since each run still takes several days on 4 GPUs. However we were able to run a GN-prox-linear until loss 3.15 which maintained a 4.9x reduction over SOAP, which we think is a good proxy for a well-tuned GN run given our results comparing the two methods (Section 6.2). (The GN-prox-linear method is much faster to run since it does not require linesearch).  We will include results for a complete set of experiments on longer iteration complexity runs in the final draft of the paper.
> [Update]: With some further tuning of the inner optimization for full GN we obtained a 5x step reduction over SOAP at loss 3.16. We have not yet completed a full hyperparameter sweep, so there may still be room for improvement, but this indicates that better inner optimization can help full GN to maintain its gains over SOAP in later stages of training. We will continue training to lower loss values and include results in the final draft of the paper.

---

### Official Review · Reviewer_hy1M · 2025-10-31

**Soundness:** 4
**Presentation:** 3
**Contribution:** 4
**Rating:** 8
**Confidence:** 4

**Summary:**

This paper investigates the upper bound potential of second-order optimization for large language models (LLMs) using full Gauss-Newton (GN) preconditioning. By implementing a memory-feasible GN method via Jacobian-vector products, the authors estimate the theoretical limit of convergence speed relative to modern optimizers like SOAP, Muon, and AdamW.
Experiments on 45M- and 150M-parameter LLaMA models show 5.4× faster convergence and larger critical batch sizes for GN compared to SOAP. The layerwise GN variant nearly matches full GN, implying that most curvature information is local and that higher-order loss terms are unnecessary.
Overall, the paper provides a rigorous and well-documented study that clarifies how far second-order optimization could improve LLM training efficiency, even if the proposed method itself is not yet practical at scale.

**Strengths:**

* Clearly defined and novel research question: measuring the upper bound of second-order optimization performance for LLMs.
* Strong empirical methodology, including detailed hyperparameter sweeps and compute transparency.
* Insightful findings: full GN drastically improves iteration efficiency, and layerwise GN nearly replicates its gains.
* Excellent documentation and reproducibility, with appendices covering training setups, tuning, and resources.
* Practical relevance: establishes a benchmark for how close current optimizers are to theoretical limits.

**Weaknesses:**

* Experiments are limited to 150M-parameter models, far smaller than modern LLMs.
* The study focuses on iteration count, not wall-clock time or energy cost, so real-world efficiency gains remain unclear.
* Comparisons may be slightly biased since the GN method uses Muon as an inner optimizer, giving it an inherent advantage.
* No multiple-run variance reporting; results could benefit from confidence intervals.
* Theoretical analysis of why layerwise GN performs nearly as well as full GN is missing.

**Questions:**

* How much slower is each GN iteration in wall-clock time compared to SOAP or Muon?
* Would the results generalize to models with stronger inter-layer dependencies (e.g., MoE architectures)?
* How sensitive are the results to the choice of inner optimizer (Muon vs. AdamW)?
* Can mixed-precision or blockwise GN approximations retain most of the benefits?
* Does the GN method still outperform others in smaller-batch or lower-data regimes?

---

> ### Author Response · Authors · 2025-11-19
>
> We thank the reviewer for their time and thoughtful comments, and we provide responses below. We will first summarize two main changes in our updated version of the paper and then address the reviewer’s specific questions:
>
> (1) We have updated the presentation of Algorithm 2 (“Linearized model method”). We renamed this algorithm to “GN-prox-linear” since the algorithm essentially takes the form of a prox-linear method as in [1,2]. We have also updated the notation in the algorithm boxes for both Algorithm 1 and 2. We hope that this clarifies some confusion on the “Linearized model method” in particular.
>
> (2) We have implemented a more efficient method for the Gauss-Newton and GN-prox-linear methods. In our original experiments there was roughly 4x overhead for these methods compared to Muon, making experiments very expensive. The new implementation requires only one additional forward pass (+some negligible computation in the logits-space for Gauss-Newton), and results in roughly 1.5x overhead. We provide details for the new implementation in Appendix B as well as in our code. While our method is still not meant to be practical given that it still relies on an iterative Muon subroutine, the more efficient implementation enables us to more easily scale experiments.
>
> In addition to the updated paper, we provide an anonymous code repository here: https://anonymous.4open.science/r/full-gauss-newton-7528.

---

> > ### Author Response · Authors · 2025-11-19
> >
> > To the reviewer’s specific concerns:
> >
> > > Experiments are limited to 150M-parameter models, far smaller than modern LLMs.
> >
> > We acknowledge that the model scale is limited; however, we still think the findings are informative for future directions of optimization methods for LLMs.
> >
> > > The study focuses on iteration count, not wall-clock time or energy cost, so real-world efficiency gains remain unclear.
> >
> > Since our optimizer is not meant to be practical we are not proposing a method to gain efficiency in wall-clock time or energy. However, we believe that the implications of potential gains in iteration complexity/the large batch size regime are valuable given the potential benefits in reducing serial runtime (eg if work can be parallelized across many GPUs). Additionally, our results in Section 6.1.2 on batch size scaling suggest that the Gauss-Newton method can retain higher sample efficiency for large batch sizes.
> >
> > > Comparisons may be slightly biased since the GN method uses Muon as an inner optimizer, giving it an inherent advantage.
> >
> > Could you clarify what you mean by inherent advantage? The inner optimizer only dictates how completely the least squares subproblem is solved in each step. (Further discussion on the choice of inner optimizer is given in response to the question below on Muon vs AdamW).
> >
> > > No multiple-run variance reporting; results could benefit from confidence intervals.
> >
> > Due to the resource limitations we do not report multiple-run variance.
> >
> > > Theoretical analysis of why layerwise GN performs nearly as well as full GN is missing.
> >
> > We leave theoretical analysis of layerwise GN to full GN as outside the scope of our work. [3] shows that for deep linear networks, both layerwise and full GN exhibit exponential convergence rates, suggesting that both of them might be comparable.
> >
> > > How much slower is each GN iteration in wall-clock time compared to SOAP or Muon?
> >
> > The original implementation had roughly 4x overhead as compared to Muon. The new implementation has roughly 1.5x overhead which makes the GN method much more feasible for experiments.
> >
> > > Would the results generalize to models with stronger inter-layer dependencies (e.g., MoE architectures)?
> >
> > This is a good question – we haven’t considered MoE architectures, but will leave this to future work!
> >
> > > How sensitive are the results to the choice of inner optimizer (Muon vs. AdamW)?
> >
> > From our preliminary experiments we found that Muon worked substantially better than AdamW as the inner optimizer. Specifically in the batch size vs loss experiments (Figure 2) the gap between Gauss-Newton and the “upper bound” line was tighter for Muon than AdamW. In general our experiments suggest that the results are fairly sensitive to the inner optimization procedure, and it is possible that with further optimization the Gauss-Newton method could achieve even better performance.
> >
> > > Can mixed-precision or blockwise GN approximations retain most of the benefits?
> >
> > Mixed-precision GN approximations are outside the scope of our paper so we will leave this to future work as well. What do you mean by blockwise? The layerwise GN method corresponds to the block diagonal GN matrix if this is what you are referring to.
> >
> > > Does the GN method still outperform others in smaller-batch or lower-data regimes?
> >
> > From our experiments in 1x Chinchilla setting comparing batch size vs loss (Figure 2), SOAP closely matches GN for small batch size. We focus on the large batch size regime due to prior work suggesting that the effect of preconditioners may not appear at small batch size [4,5].
> >
> > We thank you again for the helpful comments on our paper. We believe that we answered the main questions raised in the review, and would appreciate it if you would raise your score if you believe that your concerns were addressed.
> >
> > References:
> >
> > [1] James V. Burke. Descent methods for composite nondifferentiable optimization problems. Math. Program., 33(3): 260–279, December 1985. ISSN 0025-5610. doi: 10.1007/BF01584377. URL https://doi.org/10.1007/BF01584377.
> >
> > [2] Dmitriy Drusvyatskiy. The proximal point method revisited, 2017. URL https://arxiv.org/abs/1712.
> > 06038.
> >
> > [3] Bernacchia, Alberto, Máté Lengyel, and Guillaume Hennequin. "Exact natural gradient in deep linear networks and its application to the nonlinear case." Advances in Neural Information Processing Systems 31 (2018).
> >
> > [4] Prateek Jain, Sham M. Kakade, Rahul Kidambi, Praneeth Netrapalli, and Aaron Sidford. Accelerating stochastic gradient descent for least squares regression, 2018a. URL https://arxiv.org/abs/1704.08227.
> >
> > [5] Guodong Zhang, Lala Li, Zachary Nado, James Martens, Sushant Sachdeva, George E. Dahl, Christopher J. Shallue,and Roger Grosse. Which algorithmic choices matter at which batch sizes? insights from a noisy quadratic model, 2019. URL https://arxiv.org/abs/1907.04164.

---

### Official Review · Reviewer_Hzbf · 2025-11-01

**Soundness:** 3
**Presentation:** 2
**Contribution:** 3
**Rating:** 6
**Confidence:** 3

**Summary:**

The paper provides an empirical study of the application of the Gauss-Newton algorithm in LLM training. The paper studies a valid approximation method for lowering the memory and computational cost of the original Gauss-Newton method, making it feasible for LLM training. Empirical results show potential of this second-order approach, especially under large batch settings.

**Strengths:**

1. The motivation of the paper is interesting and inspiring, bringing second-order optimization back to the regime of LLM training, and shows promising performance.
2. The idea for approximating the full Gauss-Newton is interesting, making the algorithm practical and applicable to practical problems.

**Weaknesses:**

1. The notations and claims in the paper may need clearer explanations. I suggest the authors include more intuition on how this approximation is valid in approximating the updates of real Gauss-Newton and what the major differences are between the proposed approach and the real Gauss-Newton method.
2. For the experiment part, I have two major concerns. Firstly, as an empirical study of the application of the Gauss-Newton algorithm, the experiment scale seems not large enough, since the paper only examines models of size 45M and 150M. Also, it seems that only when a very large batch size (compared to the full token number) is adopted, the benefits of the Gauss-Newton method are clear, which seems somewhat limited.

**Questions:**

1. In practice, I am curious about how we should tune the hyperparameter $ N $ (the inner loop length)? Schedule choices may also be a concerning point. There seems to be still a long way to go to put this second-order algorithm into practical training settings.
2. What about the wall-clock training time of the algorithm? How is it compared to Muon or Adam?

---

> ### Author Response · Authors · 2025-11-19
>
> We thank the reviewer for their time and thoughtful comments, and we provide responses below. We will first summarize two main changes in our updated version of the paper and then address the reviewer’s specific questions:
>
> (1) We have updated the presentation of Algorithm 2 (“Linearized model method”). We renamed this algorithm to “GN-prox-linear” since the algorithm essentially takes the form of a prox-linear method as in [1,2]. We have also updated the notation in the algorithm boxes for both Algorithm 1 and 2. We hope that this clarifies some confusion on the “Linearized model method” in particular.
>
> (2) We have implemented a more efficient method for the Gauss-Newton and GN-prox-linear methods. In our original experiments there was roughly 4x overhead for these methods compared to Muon, making experiments very expensive. The new implementation requires only one additional forward pass (+some negligible computation in the logits-space for Gauss-Newton), and results in roughly 1.5x overhead. We provide details for the new implementation in Appendix B as well as in our code. While our method is still not meant to be practical given that it still relies on an iterative Muon subroutine, the more efficient implementation enables us to more easily scale experiments.
>
> In addition to the updated paper, we provide an anonymous code repository here: https://anonymous.4open.science/r/full-gauss-newton-7528.

---

> > ### Author Response · Authors · 2025-11-19
> >
> > To the reviewer’s specific concerns:
> >
> > > The notations and claims in the paper may need clearer explanations. I suggest the authors include more intuition on how this approximation is valid in approximating the updates of real Gauss-Newton and what the major differences are between the proposed approach and the real Gauss-Newton method.
> >
> > The equivalence between the solution of the subproblem we solve and the actual Gauss-Newton method is provided in Appendix A. Our method follows from Hessian-free methods that solve an incomplete (unconverged) optimization of this subproblem rather than storing an approximation to the Hessian [3]. Additionally we have updated much of the notation in the new version which we hope is more clear.
> >
> > > For the experiment part, I have two major concerns. Firstly, as an empirical study of the application of the Gauss-Newton algorithm, the experiment scale seems not large enough, since the paper only examines models of size 45M and 150M. Also, it seems that only when a very large batch size (compared to the full token number) is adopted, the benefits of the Gauss-Newton method are clear, which seems somewhat limited.
> >
> > We acknowledge that the model scale is limited; however, we still think the findings are informative for future directions of optimization methods for LLMs. We focus on the large batch size regime due to prior work suggesting that the effect of preconditioners may not appear at small batch size [4,5]. We think that benefits in the large batch size regime are valuable given the potential benefits in reducing serial runtime (eg if work can be parallelized across many GPUs).
> >
> > > In practice, I am curious about how we should tune the hyperparameter  (the inner loop length)? Schedule choices may also be a concerning point. There seems to be still a long way to go to put this second-order algorithm into practical training settings.
> >
> > We would like to clarify that we are not proposing that our second-order algorithm be put into practical training settings; rather, it is meant to serve as an empirical study aimed at understanding performance limits of second-order optimizers. In our experiments we use the inner loop length to vary the effective batch size of the Gauss-Newton method (batch size referring to how much data is seen per one parameter update on the actual model). As to scheduling choices, we tried three learning rate schedules which are illustrated in Appendix D.  In our experiments we found that the global cosine schedule outperforms the global+inner cosine schedule at small to medium batch size, but the constant+inner cosine schedule can be helpful for runs at large batch size for Gauss-Newton.
> >
> > > What about the wall-clock training time of the algorithm? How is it compared to Muon or Adam?
> >
> > We would like to again emphasize that we are not proposing that our second-order algorithm as a practical algorithm – however as to wall clock time, with our new implementation there is only a 1.5x overhead as compared to Adam.
> >
> > We thank you again for the helpful comments on our paper. We believe that we answered the main drawbacks raised in the review, and would appreciate it if you would raise your score if you believe that your concerns were addressed.
> >
> > References:
> >
> > [1] James V. Burke. Descent methods for composite nondifferentiable optimization problems. Math. Program., 33(3): 260–279, December 1985. ISSN 0025-5610. doi: 10.1007/BF01584377. URL https://doi.org/10.1007/BF01584377.
> >
> > [2] Dmitriy Drusvyatskiy. The proximal point method revisited, 2017. URL https://arxiv.org/abs/1712.
> > 06038.
> >
> > [3] James Martens. Deep learning via hessian-free optimization. In Proceedings of the 27th International Conference on International Conference on Machine Learning, ICML’10, pp. 735–742, Madison, WI, USA, 2010. Omnipress. ISBN 9781605589077.
> >
> > [4] Prateek Jain, Sham M. Kakade, Rahul Kidambi, Praneeth Netrapalli, and Aaron Sidford. Accelerating stochastic gradient descent for least squares regression, 2018a. URL https://arxiv.org/abs/1704.08227.
> >
> > [5] Guodong Zhang, Lala Li, Zachary Nado, James Martens, Sushant Sachdeva, George E. Dahl, Christopher J. Shallue,and Roger Grosse. Which algorithmic choices matter at which batch sizes? insights from a noisy quadratic model, 2019. URL https://arxiv.org/abs/1907.04164.

---

### Official Review · Reviewer_CWUm · 2025-11-02

**Soundness:** 4
**Presentation:** 4
**Contribution:** 4
**Rating:** 8
**Confidence:** 5

**Summary:**

The paper test the performance of full 2nd-order method (Gauss-Newton) on LLM pretraining. The aim of the paper is to explore the potential of classical 2nd-order methods in mordern pretrain tasks.  Although the proposed implementation is still compute intensive, this trial is meaningful and provides valuable guidance for the future optimizer design.

**Strengths:**

The topic of "exploring the 2nd-order methods in LLM pretrain" is crucially important. The presentation is clear. The experiments are sound. The message is valuable and quite worth sharing with the community.

**Weaknesses:**

The presentation can be further improved, as I explained below.

**Questions:**

**Summary:** The paper test the performance of full 2nd-order method (Gauss-Newton) on LLM pretraining. The aim of the paper is to explore the potential of classical 2nd-order methods in mordern pretrain tasks.  Although the proposed implementation is still compute intensive, this trial is meaningful and provides valuable guidance for the future optimizer design.

**Strength:**  The topic of "exploring the 2nd-order methods in LLM pretrain" is crucially important. The experiments are sound. The message is valuable and quite worth sharing with the community.

**Disclaimer:** Many may criticize that the computational cost of this paper remains high, which is true. However, I do not consider it a critical issue. For me, the directional guidance provided by this work outweighs its computational limitations. The reasons are as follows:

1. Although traditional second-order optimization methods have a rich theoretical foundation, there have been almost no meaningful attempts to apply them directly to modern neural networks or large language models, except a few early attempts on RNNs 10 years ago by Martins and Sustkever.  For modern LLM pretrain, we only know that several heavily approximated versions of Gauss-Newton can work in practice (e.g., Muon and Adam), but it remains unclear whether the non-approximated versions would be effective. This script gives a postive answer on a meaningful setting (nanoGPT speedrun).

2. Approximated methods often develop new characteristics that lead to new theoretical interpretations—for instance, Muon exhibits properties related to matrix orthogonalization. Researchers tend to construct new theoretical frameworks to explain these methods rather than treating them as approximations of classical approaches. While this promotes academic diversity and inspire new thoughts, it also creates confusion in algorithm design: Should we follow classical theories or newer principles? This paper demonstrates that following classical theory can indeed lead to success. This directional guidance is highly valuable, and I believe this work has the potential to inspire many future optimizer designs.

In summary, I think this is an important paper that provides important directional guidance to future optimizer design. The computational overhead is a minor issue since the main contribution lies in "guidance" rather than "an algorithm itself".  I vote for clear acceptance.

**Questions related to presentation:**  The paper presentation is not prefect and can be further improved.

**Question 1:**
 I am quite confused by Section 7.2. LINEARIZED MODEL METHOD.

1-1. What is the explicit form of this Algorithm 2? As an analogy to previous sections: the minimization over 2nd-order Taylor of the loss over the linearized model gives  w= w - G-1 grad, does algorithm 2 admit closed-form expression, and what is the closed-form if it exists?

1-2. Is algorithm 2 still a 2nd-order algorithm?

1-3. What is the implication on the loss landscape if Algorithm 2 works well?

1-4. How do you solve the subproblem in Section 7.2.

1-5. The name "Algorithm 2: Linearized model method" is quite misleading. Both Algorithm 1 and 2 are based on the Linearized model, an  if I understand correctly, Algorithm 2 actually incorporates higher-order loss function information than Algorithm 1.  The name "Linearized model method" may leave the impression that "Algorithm 1 incorporates higher-order information than Algorithm 2", but actually it is the other way around (correct me if wrong).


**Question 2:** The algorithm presentation (Algorithm 1 and Algorithm 2) is confusing. It is impossible to understand the line "Update", e.g., in line 231. I suggest adding a hyperlink to tell readers which equation is used in defining this "Update" operator. Similarly, the line "Line search" is confusing as well. In optimization theory, line search usually refers to a particular procedure of choosing the stepsize of GD. But here you mean something different. Please add more detailed description.


**Question 3:** "We start all runs from the same AdamW post-warmup checkpoint." I dont understand this sentence. What do you mean by "all runs" and what is "AdamW post-warmup checkpoint"? Further, why not use random initialization?

**Questions related to experiments:**

**Question 4:** When computing L(θ_t + α(θ̂ − θ_t)) during line search, do you reuse the same data as in the inner loop, a held-out/proxy mini-batch, or freshly sampled data? Please specify for both full GN and layerwise GN

**Question 5:** Ablation study on line search. I’m wondering whether other optimization algorithms also rely on line search. If they don’t, why does line search improve the performance of Gauss–Newton? In addition, how much of Gauss–Newton’s performance gain comes from the algorithm itself rather than from the use of line search?

**Question 6:** The current script uses  GN^-1: w = w  - G^-1 grad. How about GN^-1/2: w = w  - G^-1/2 grad? Is there any feasible way to implement GN^-1/2? How would you expect it to perform?

**Question 7:** How would GN perform if you keep training on more tokens? Will the benefit shrink as the token size reaches 1x or 2x Chinchilla, or even more tokens?

**Question 8:**  In Figure 1, the x-axis is the number of outer steps, while Gauss–Newton uses a much larger batch size (240 M tokens vs. 40 M for AdamW/Muon) and each “step” includes several inner optimizer iterations and line-search evaluations. This makes the comparison on a per-step basis potentially misleading. I suggest reporting loss versus total tokens processed.

**Code:** No code is provided in the submission. I strongly suggest the authors provide an anonymous code link.  I will check the code, and I believe it is the most efficient way to help address my questions above.

I am more than happy to raise my score to 10 if my questions are addressed.

---

> ### Author Response · Authors · 2025-11-19
>
> We thank the reviewer for their time and thoughtful comments, and we provide responses below. We will first summarize two main changes in our updated version of the paper and then address the reviewer’s specific questions:
>
> (1) We have updated the presentation of Algorithm 2 (“Linearized model method”). We renamed this algorithm to “GN-prox-linear” since the algorithm essentially takes the form of a prox-linear method as in [1,2]. We have also updated the notation in the algorithm boxes for both Algorithm 1 and 2. We hope that this clarifies some confusion on the “Linearized model method” in particular.
>
> (2) We have implemented a more efficient method for the Gauss-Newton and GN-prox-linear methods. In our original experiments there was roughly 4x overhead for these methods compared to Muon, making experiments very expensive. The new implementation requires only one additional forward pass (+some negligible computation in the logits-space for Gauss-Newton), and results in roughly 1.5x overhead. We provide details for the new implementation in Appendix B as well as in our code. While our method is still not meant to be practical given that it still relies on an iterative Muon subroutine, the more efficient implementation enables us to more easily scale experiments.
>
> In addition to the updated paper, we provide an anonymous code repository here: https://anonymous.4open.science/r/full-gauss-newton-7528.

---

> > ### Author Response · Authors · 2025-11-19
> >
> > To the reviewer’s specific concerns:
> >
> > **Question 1**: We have renamed the “Linearized model method” to “GN-prox-linear” and hopefully this solves some of the confusion on this method. The subproblem here is solved in the same way as the Gauss-Newton subproblem, with Muon optimizing the loss on the linearized model. (For Gauss Newton, Muon is optimizing the quadratic approximation of the loss on the linearized model). This does not explicitly fall in the class of 2nd order algorithms as it uses higher order loss terms. However, this is still a convex subproblem. Given that this method works well, one implication is that another way of thinking about neural network optimization could be in terms of solving a sequence of kernel classification tasks. Please let us know if you have any remaining questions on this method.
> >
> > **Question 2**: The algorithm boxes have been updated in the new version so that both algorithms are framed in terms of the subproblem rather than adding the details of the specific subroutine we use. Details on how the subproblems are solved is given in Section 5. For the line search we evaluate the true model loss on new data on step sizes 2^{-i/2} for i in {0,...,4} (See Appendix D.2).
> >
> > **Question 3**: We use the AdamW warmup as a general warmup phase to avoid defining a warmup schedule for the GN methods (which by default use step size 1) and to make sure all runs are well-initialized. It is possible that random initialization could be okay but likely the Gauss-Newton would need some sort of warmup to maintain stability.
> >
> > **Question 4**: In our experiments we use freshly sampled data for the line search for both GN and layerwise GN.
> >
> > **Question 5**: From our experiments, the GN-prox-linear method works well without linesearch while GN seems to get the direction well but needs the line search for step size. As to why linesearch helps GN,  [3] shows that G^-1 step sizes could lead to suboptimal performance without linesearch. We can measure the directional alignment between GN-prox-linear, GN, and SOAP to investigate this further.
> >
> > **Question 6**: We are not sure how to get G^{-½} out of our current framework. In general, we would expect it to not outperform GN^{-1} at sufficiently large batch size based on the findings of [3], but we aren’t positive.
> >
> > **Question 7**: We will launch some longer runs to investigate this. (Our new implementation should allow us to scale the amount of data past 1x Chinchilla, this was infeasible previously).
> >
> > **Question 8**: The idea for Figure 1 is to measure iteration complexity, putting each optimizer in the regime where further increasing batch size gives negligible benefit. For Adam and Muon this seems to be around 40M given our CBS experiments, so to save on compute we did not scale the batch size to 240M. We did use 240M batch size for SOAP. If you would find it useful, we can run Adam and Muon with a larger batch size to confirm that there are negligible gains. We consider sample efficiency in Figure 2.
> >
> > **Code**: Here is our anonymized code repository https://anonymous.4open.science/r/full-gauss-newton-7528.
> >
> > We thank you again for the helpful comments on our paper. We believe that we answered the main concerns raised in the review, and would appreciate it if you would raise your score if you believe that your concerns were addressed.
> >
> > References:
> >
> > [1] James V. Burke. Descent methods for composite nondifferentiable optimization problems. Math. Program., 33(3): 260–279, December 1985. ISSN 0025-5610. doi: 10.1007/BF01584377. URL https://doi.org/10.1007/BF01584377.
> >
> > [2] Dmitriy Drusvyatskiy. The proximal point method revisited, 2017. URL https://arxiv.org/abs/1712.
> > 06038.
> >
> > [3] Liu, Bingbin, et al. "Adam or Gauss-Newton? A Comparative Study In Terms of Basis Alignment and SGD Noise." arXiv preprint arXiv:2510.13680 (2025).

---

> > > ### Comment · Reviewer_CWUm · 2025-11-26
> > > **A follow-up question**
> > >
> > > Thanks for the rebuttal! It is always nice to see that the overhead is reduced from 4x to 1.5x.
> > >
> > > I have a follow-up question regarding the overhead: In your refined implementation (the 1.5x overhead version), what proportion of the total wall-clock time is spent in each of the main components, e.g., forward pass, JVPs, backward pass, the Muon optimizer in the inner loop, and the line search?
> > >
> > > It would be nice to have a table to illustrate this.

---

> > > > ### Author Response · Authors · 2025-12-03
> > > >
> > > > Thanks for the question! Here is a time breakdown of one training step on a 45M model on a batch size of 4 sequences, using 3 inner steps and 5 linesearch step size candidates. The timing is given with JIT disabled, but in our code we JIT the inner train step which makes the entire inner loop take **0.1027s** here. The "calls" column is given as number of calls per parent call.
> > > >
> > > > ### Timing breakdown for GN train step
> > > >
> > > > | Section              | Calls | Total (s) | Mean (s) |
> > > > |----------------------|-------|-----------|----------|
> > > > | **inner_loop**       | 1     | 4.4046    | 4.4046   |
> > > > | &nbsp;&nbsp;inner_step | 3     | 4.1979    | 1.3993   |
> > > > | &nbsp;&nbsp;&nbsp;&nbsp;gn_value_and_grad | 1 | 0.8805 | 0.8805 |
> > > > | &nbsp;&nbsp;&nbsp;&nbsp;&nbsp;&nbsp;gn_linearize | 1 | 0.2783 | 0.2783 |
> > > > | &nbsp;&nbsp;&nbsp;&nbsp;&nbsp;&nbsp;gn_jvp_Jv | 1 | 0.1256 | 0.1256 |
> > > > | &nbsp;&nbsp;&nbsp;&nbsp;&nbsp;&nbsp;gn_grad_Ly | 1 | 0.0573 | 0.0573 |
> > > > | &nbsp;&nbsp;&nbsp;&nbsp;&nbsp;&nbsp;gn_Hv_Jv | 1 | 0.0671 | 0.0671 |
> > > > | &nbsp;&nbsp;&nbsp;&nbsp;&nbsp;&nbsp;gn_Jt_pullback | 1 | 0.3016 | 0.3016 |
> > > > | &nbsp;&nbsp;&nbsp;&nbsp;&nbsp;&nbsp;gn_quadratic_loss | 1 | 0.0458 | 0.0458 |
> > > > | &nbsp;&nbsp;&nbsp;&nbsp;gn_opt_step | 1 | 0.4447 | 0.4447 |
> > > > | **linesearch**       | 1     | 2.9851    | 2.9851   |
> > > > | &nbsp;&nbsp;ls_candidate | 5 | 2.8995 | 0.5799 |

---

### Meta-Review · Area_Chair_B7sT · 2025-12-26

**Summary:**

This paper studies second-order method for LLM training. It is inspired by the success of several algorithms for LLM that utilizes second-order information such Shampoo, SOAP, and Muon. However, these algorithms don’t fully utilize second-order information, and in the literature there lacks a systematic study on this. This paper conducts a study on the full Gauss-Newton (GN) for LLM, and the results can potentially guide future studies of second-order algorithms for LLM. The authors clearly explained their motivation, and conducted extensive numerical experiments. Overall this is a nice contribution to LLM algorithms and will shed light on designing efficient second-order algorithms for LLM.

**Reviewer Concerns:**

Addressed: more experimental results. Clarify in some places.

**Reviewer Scores:**

Overall, this paper received high scores from three reviewers. Given the thorough rebuttal, I expect that some reviewers might increase their scores if discussions were enabled.

---

### Decision · Program_Chairs · 2026-01-26

Accept (Poster)